Identification and validation of shared gene signature of kidney renal clear cell carcinoma and COVID-19

Nie Jianqiang 1
Yang Hailang 1
Liu Xiaoqiang 1 2
Deng Wen 1 2 urodeng@126.com
Fu Bin 1 2 urofbin@163.com
1 First Affiliated Hospital of Nanchang University , Nanchang , China
2 Jiangxi Institute of Urology , Nanchang , China
Basharat Zarrin
Electronic publication date: 2024 Mar 4
Publication date: 2024
Volume: 12
Electronic Location ID: e16927
Received 2023 Aug 30; Accepted 2024 Jan 22
Copyright: © 2024 Nie et al.
Copyright year: 2024
Copyright holder: Nie et al.
License: This is an open access article distributed under the terms of the Creative Commons Attribution License, which permits unrestricted use, distribution, reproduction and adaptation in any medium and for any purpose provided that it is properly attributed. For attribution, the original author(s), title, publication source (PeerJ) and either DOI or URL of the article must be cited.
License URL: https://creativecommons.org/licenses/by/4.0/

Keywords: KIRC, COVID-19, Shared gene, Comorbidity, Co-pathogenesis

Funding: National Natural Science Foundation of P.R. China 81560419 This study was supported by the National Natural Science Foundation of P.R. China (Grant Nos. 81560419). The funders had no role in study design, data collection and analysis, decision to publish, or preparation of the manuscript.

==============================
Background

COVID-19 is a severe infectious disease caused by the SARS-CoV-2 virus, and previous studies have shown that patients with kidney renal clear cell carcinoma (KIRC) are more susceptible to SARS-CoV-2 infection than the general population. Nevertheless, their co-pathogenesis remains incompletely elucidated.

Methods

We obtained shared genes between these two diseases based on public datasets, constructed a prognostic risk model consisting of hub genes, and validated the accuracy of the model using internal and external validation sets. We further analyzed the immune landscape of the prognostic risk model, investigated the biological functions of the hub genes, and detected their expression in renal cell carcinoma cells using qPCR. Finally, we searched the candidate drugs associated with hub gene-related targets from DSigDB and CellMiner databases.

Results

We obtained 156 shared genes between KIRC and COVID-19 and constructed a prognostic risk model consisting of four hub genes. Both shared genes and hub genes were highly enriched in immune-related functions and pathways. Hub genes were significantly overexpressed in COVID-19 and KIRC. ROC curves, nomograms, etc., showed the reliability and robustness of the risk model, which was validated in both internal and external datasets. Moreover, patients in the high-risk group showed a higher proportion of immune cells, higher expression of immune checkpoint genes, and more active immune-related functions. Finally, we identified promising drugs for COVID-19 and KIRC, such as etoposide, fulvestrant, and topotecan.

Conclusion

This study identified and validated four shared genes for KIRC and COVID-19. These genes are associated with immune functions and may serve as potential prognostic biomarkers for KIRC. The shared pathways and genes may provide new insights for further mechanistic research and treatment of comorbidities.

Introduction

COVID-19 is a novel pneumonia caused by the severe acute respiratory syndrome coronavirus 2 (SARS-CoV-2) (Zhu et al., 2020). Since December 2019, it has resulted in significant loss of life and caused global economic and physical distress (McKee & Stuckler, 2020). According to Johns Hopkins University data as of January 10, 2023, there have been over 660 million global infections and more than 6.7 million deaths (Johns Hopkins University and Medicine, 2023). Research suggests that individuals with compromised immune systems are more vulnerable to SARS-CoV-2 infection (Grasselli et al., 2020). Cancer patients face an elevated risk of COVID-19 infection and are prone to quicker onset of complications and deterioration due to potential immune system suppression caused by cancer and its treatments (Wang et al., 2020).

Despite progress in renal cell carcinoma (RCC) diagnosis and management, it continues to be one of the most lethal urological malignancies (Cairns, 2010). Globally, approximately 431,000 new cases of kidney cancer are diagnosed annually, resulting in over 179,000 deaths and posing a significant global public health burden (Sung et al., 2021). Kidney renal clear cell carcinoma (KIRC) is the predominant type, comprising approximately 70% of RCC (Zhang, Zeng & Hu, 2020). The kidney is a major target organ for SARS-CoV-2 infection, and COVID-19 frequently leads to acute tubular necrosis, a common complication (Kellum, van Till & Mulligan, 2020). ACE2 serves as the receptor for SARS-CoV-2 to enter host cells (Weiss & Navas-Martin, 2005). Additionally, NPR1, DPP4, ANPEP, ENPEP, and TMPRSS2 have been identified as co-receptors or cofactors for initiating SARS-CoV-2 infection (Choong et al., 2023; Qi et al., 2020; Tang et al., 2021; Hossain, Akter & Uddin, 2021). According to Choong et al. (2023), RCC cells retain the crucial receptors and cofactors required for SARS-CoV-2 viral entry. Tripathi et al. (2020) revealed significant overexpression of these genes in KIRC, suggesting their potential involvement in regulating cellular immunity and immune infiltration. However, the shared underlying pathogenic mechanisms have not been thoroughly investigated.

The rapid progress in next-generation sequencing technology offers an effective approach to exploring the shared pathogenic mechanisms underlying different diseases (Kilpinen & Barrett, 2013). Analyzing the shared transcriptional characteristics of KIRC and COVID-19 may yield novel insights into the pathogenesis of both conditions. In this study, we collected datasets from TCGA, GEO, and ICGC databases for both diseases. Using bioinformatics analysis, we constructed a risk model comprising four hub genes. We validated this model using both internal and external validation sets and investigated its role in immune infiltration. The objective of this research is to aid in understanding the shared pathogenic mechanisms of COVID-19 and KIRC among researchers and offer valuable recommendations for the clinical management of patients with these concurrent conditions.

Materials and Methods

Data sources and preprocessing

We obtained the expression profiles of COVID-19 patients from the GEO database. Two cohorts, namely GSE196822 (GPL20301) and GSE211979 (GPL16791) consisting of a total of 47 samples (33 patient samples and 14 normal samples), were chosen. If multiple probes matched a single gene, the gene’s average expression was calculated. The study encompassed a total of 22,235 genes. We employed the “limma” R package to identify differentially expressed genes (DEGs) between normal and COVID-19 patient samples. These DEGs were named COVID-19-DEGs and filtered based on logFC ≥1 and adjusted p-value < 0.05.

We downloaded the expression profiles of KIRC patients from the TCGA database KIRC cohort (downloaded on 2023.01.03), which comprised 542 tumor tissue samples and 72 adjacent noncancerous tissue samples. If a gene had multiple probes, we calculated the average expression of that gene. A total of 60,550 genes were acquired. We utilized the “limma” R package to obtain DEGs between tumor and normal groups. These DEGs were named KIRC-DEGs and were filtered based on logFC ≥ 1.5 and FDR < 0.05. We also downloaded the corresponding mutation data and clinical information for KIRC patients from the TCGA database. We excluded samples with a survival time of 0 or missing information. Ultimately, we selected a final cohort of 533 patients for analysis.

To validate the prognostic risk model, we acquired the ICGC-RECA-EU cohort from the ICGC database and selected 91 KIRC patients with reliable prognostic information for external validation.

Identification and functional annotation of shared genes

To identify the most relevant genes associated with COVID-19, we employed weighted correlation network analysis (WGCNA) to classify the COVID-19-DEGs into distinct modules. We considered the module genes significantly associated with COVID-19 as candidate genes. The shared genes were obtained by intersecting these candidate genes with the KIRC-DEGs. To perform function enrichment analysis on the shared genes, we utilized the “clusterProfiler” R package, focusing on Kyoto Encyclopedia of Genes and Genomes (KEGG) and Gene Ontology (GO).

Machine learning-based variable screening and model construction

The shared genes in the TCGA-KIRC training set were analyzed by univariate Cox regression to select the significant shared genes (p < 0.05), followed by Lasso-Cox regression analysis, which is a novel algorithm for screening parameters in high-dimensional data (Goeman, 2010). Through Lasso-Cox regression analysis, we further reduced the number of shared genes. Finally, using multivariate Cox regression analysis, we narrowed down the genes and constructed a risk model consisting of hub genes, which represent the signature genes of the model.

The risk score is calculated using the following formula:

RiskScore=∑i=0n⁡(βi∗xi)

where n is the number of genes, β is the coefficient and x is the expression level of the gene.

Validation of hub genes expression

Through the previous analysis, we identified differential expression of hub genes in both COVID-19 and KIRC. To further validate our findings, we examined the expression of these hub genes in RCC cells.

Cell culture

RCC cell lines (ACHN, A498, Caki-1, OS-RC-2, and 786-O) and renal tubular epithelial cells (HK-2) were obtained from the Kunming Cell Bank of the Chinese Academy of Sciences Typical Culture Collection Committee and stored in liquid nitrogen. The cells were cultured in a CO2 incubator at 37 °C.

Real-time fluorescent quantitative PCR amplification (qPCR)

Total RNA was extracted from cells of the above cell lines during the logarithmic growth phase. The extracted RNA was reverse-transcribed into cDNA using a reverse transcription kit provided by Tiangen Company (Beijing, China). Fluorescence expression levels were measured using a NanoDrop™ 2000 spectrophotometer (Thermo Fisher Scientific, Waltham, MA, USA). Amplification was conducted using a Bio-Rad PCR machine (Bio-Rad, Shanghai, China), and β-actin was utilized for normalization. The relative expression level of the gene was calculated using the 2−ΔΔCt method.

Assessment of prognostic features and validated predictive power

The TCGA-KIRC training set was divided into low- and high-risk groups based on the median value of the hub genetic risk score. The survival rate between these groups was assessed using the Kaplan-Meier method and log-rank test with the “survminer” R package. Receiver operating characteristic (ROC) curves were generated using the “survivalROC” R package, and the area under the curve (AUC) was calculated to evaluate the predictive power of prognostic characteristics and other clinical parameters for patients’ overall survival (OS) at 1, 3, and 5 years. Furthermore, the accuracy and robustness of the risk model were confirmed in the TCGA-KIRC testing set, the complete TCGA-KIRC cohort, and the external validation set (ICGC-RECA-EU cohort).

Principal component analysis and development and validation of prediction models for nomograms

The discriminative capability of the risk model between low- and high-risk patients was assessed by generating Principal Component Analysis (PCA) scatterplots using the “scatterplot3d” R package. Clinical parameters, including gender, age, histological grade, pathological stage, and risk score, were incorporated to develop predictive nomograms for estimating the 1-year, 3-year, and 5-year survival rates of KIRC patients in the training set, testing set, entire set, and external validation set using the “rms” R package. Calibration plots were generated to evaluate the predictive performance of prognostic nomograms for patients’ OS at 1-year, 3-year, and 5-year.

Immune landscapes for risk model

The "ESTIMATE" algorithm was employed to evaluate the tumor microenvironment and determine the immune score, stromal score, and estimated score for each tumor sample using gene expression values from the entire TCGA-KIRC cohort (Yoshihara et al., 2013) Subsequently, we examined the profiles of 22 tumor-infiltrating immune cells using the “CIBERSORT” R package and determined their relative proportions in the entire TCGA-KIRC cohort (p < 0.05) (Newman et al., 2015). Additionally, we analyzed the associations between the risk model and immune function, as well as immune checkpoints.

Relationship between the risk model and tumor mutation burden

Tumor Mutational Burden (TMB) is a potential biomarker used to predict the response to immunotherapy, and patients with a high TMB often experience greater benefits from immunotherapy (Goodman et al., 2017; Yin et al., 2020). Mutation data were obtained from the TCGA-KIRC cohort. The TMB score was calculated using the formula: (total mutations/total covered bases) × 106. We compared the TMB scores of the low- and high-risk groups and evaluated the survival differences between risk scores and TMB scores.

Construction of hub gene-based protein-protein interaction network and gene set enrichment analysis

We constructed a protein-protein interaction (PPI) network based on hub genes and their interacting genes using the GeneMANIA portal (https://genemania.org). This network was used to predict correlations among co-localization, co-expression, shared protein structural domains, and common pathways (Warde-Farley et al., 2010). Gene Set Enrichment Analysis (GSEA) is a visual method used to determine statistically significant and consistent differences in gene sets among different functional phenotypes (Subramanian et al., 2005). By comparing the enriched biological processes in the low and high-risk groups of the entire TCGA-KIRC cohort, we determined the potential biological functions of hub genes. The reference gene collection used in the GSEA was downloaded from the Molecular Signature Database (MSigDB), including c2.cp.kegg.v6.0.symbols.gmt and c5.go.v7.4.symbols.gmt. The critical criterion was set at p < 0.05.

Drug screening based on hub genes

To facilitate therapeutic development, we explored potential drug molecules significantly associated with hub genes using the Drug Signature Database (DSigDB) and cellMiner Database (Yoo et al., 2015; Shankavaram et al., 2009).

Statistical analysis

Statistical analyses were conducted using R software version 4.2.2 (R Core Team, 2022). Survival comparisons were performed using the Kaplan-Meier method and log-rank test. A significance level of p < 0.05 was used unless otherwise stated.

Results

Identification and functional annotation of shared genes

Initially, we conducted a differential expression analysis on the COVID-19 dataset to identify genes with differential expression (COVID-19-DEGs), based on logFC ≥ 1 and an adjusted p-value < 0.05 (Fig. 1A). The COVID-19-DEGs were then partitioned into eight modules using WGCNA. The modules MEbrown, MEblue, MEpurple, MEblack, MEpink, and MEgrey, which demonstrated higher module significance (MS) indicating a stronger correlation with the disease, were selected for further analysis. These modules collectively comprised 556 genes (Fig. 1B). Subsequently, we performed an intersection between these genes and differentially expressed genes in kidney renal clear cell carcinoma (KIRC-DEGs) (n = 3,106) to identify shared genes (Figs. 1C and 1D). Heatmaps of the COVID-19-DEGs and KIRC-DEGs are presented in Figs. S1 and S2, respectively, while Fig. S3 illustrates a segment of the WGCNA analysis process.

Figure 1 Identification and functional annotation of shared genes.

(A) Volcano plot illustrating differential genes in COVID-19. (B) WGCNA analysis displaying the COVID-19 associated module. (C) Volcano plot showing differential genes in KIRC. (D) Venn plot demonstrating the shared genes between COVID-19 and KIRC. (E) GO enrichment analysis of the shared genes. (F) KEGG analysis of the shared genes.

Next, we conducted an enrichment analysis of the shared genes using KEGG and GO enrichment analysis. The GO enrichment analysis revealed predominant enrichment of immune-related functions in these genes, including phagocytosis, complement activation, immunoglobulin complexes, antigen binding, and immunoglobulin receptor binding, among others (Fig. 1E). The KEGG analysis indicated that the enriched pathway associated with these genes was the cell cycle (Fig. 1F). These findings suggest a close association between the shared genes and immune function, implying that immune-related functions are likely to be impacted during the onset of the disease.

Construction of the risk model

A random assignment was performed on the entire TCGA-KIRC cohort with a 1:1 ratio, followed by univariate Cox and Lasso-Cox regression analyses conducted on 156 genes within the training set (n = 265). Initially, univariate Cox regression analysis was employed to explore the association between the expression levels of selected genes in the training set and OS time. Utilizing a significance threshold of p < 0.05 in Cox analysis, we identified 83 genes as potential risk factors associated with OS (Table S1). Subsequently, this gene set underwent refinement through the calculation of regression coefficients using the LASSO regression algorithm (Figs. 2A and 2B). This process identified the nine most valuable predictive genes (Table S2), from which a risk scoring system was established using the formula described earlier. Further refinement of these nine genes was conducted through multivariate Cox regression analysis. Ultimately, we identified four hub genes based on a Cox p value of < 0.01, namely GTSE1, CEACAM4, HECW2, and KCNMA1, as features for constructing the risk model (Fig. 2C). Specifically, GTSE1 and CEACAM4 were identified as risk factors (HR > 1), whereas HECW2 and KCNMA1 were characterized as protective factors. The risk score for each KIRC patient could be calculated by utilizing the expression values of these four genes along with the coefficients obtained from the multivariate Cox regression, employing the following formula:

Figure 2 Construction of risk models and validation in TCGA training set.

(A and B) Lasso-Cox regression analysis for gene screening. (C) Forest plot presenting hub genes obtained from multivariate analysis. (D–F) Validation of the risk model in the TCGA training set (KM curve, risk plot, ROC curve).

Risk Score = (0.786708751 * GTSE1 exp) + (0.716034712 * CEACAM4 exp) +

(−0.489951732 * HECW2 exp) + (−0.414148249 * KCNMA1 exp).

Validation of the risk model

The Risk Score was individually computed for each sample in the TCGA training set (n = 265) using the gene expression data and the risk score formula. Subsequently, the samples were divided into low and high-risk groups based on the median risk scores to evaluate the predictive performance of the risk model. Kaplan-Meier curves demonstrated a significantly improved prognosis for low-risk patients, with higher risk scores corresponding to shorter survival time and increased mortality rates (Fig. 2D). The effectiveness of the risk model in predicting patient survival at 1, 3, and 5 years was evaluated using ROC curves, revealing AUC values of 0.791, 0.755, and 0.782 for the respective time points (Figs. 2E and 2F). Survival analysis of the TCGA-KIRC testing set demonstrated a markedly worse prognosis for high-risk patients (p < 0.001), with corresponding AUC values of 0.698, 0.667, and 0.669 for 1, 3, and 5 years, respectively (Figs. 3A–3C). Comparable findings were observed across the entire TCGA-KIRC cohort and the external validation set (Figs. 3D–3I). Additionally, the distributions of risk scores, survival status, and gene expression of the model genes for each dataset were presented, further substantiating the robustness and accuracy of the risk model. The baseline characteristics for each dataset are provided in Table 1.

Figure 3 Validation of models in TCGA-KIRC testing set and entire TCGA-KIRC cohort and external independent cohort.

(A–C) Validation of the risk model in the TCGA-KIRC testing set (KM curve, ROC curve, risk plot). (D–F) Validation of the risk model in the entire TCGA-KIRC cohort (KM curve, ROC curve, risk plot). (G–I) Validation of the risk model in the external validation set ICGC-RECA-EU cohort (KM curve, ROC curve, risk plot).

Table 1 Baseline for each dataset.

	Type	TCGA-KIRC	ICGC-RECA-EU	
Entire set	Testing set	Training set	p-value	
Age	<=65	348 (65.78%)	165 (62.5%)	183 (69.06%)	0.1342	63 (69.2%)	
	>65	181 (34.22%)	99 (37.5%)	82 (30.94%)		28 (30.7%)	
Gender	FEMALE	185 (34.97%)	101 (38.26%)	84 (31.7%)	0.1361	39 (42.8%)	
	MALE	344 (65.03%)	163 (61.74%)	181 (68.3%)		52 (57.1%)	
T	T1	269 (50.85%)	139 (52.65%)	130 (49.06%)	0.7243	54 (59.3%)	
	T2	69 (13.04%)	36 (13.64%)	33 (12.45%)		13 (14.3%)	
	T3	180 (34.03%)	84 (31.82%)	96 (36.23%)		22 (24.2%)	
	T4	11 (2.08%)	5 (1.89%)	6 (2.26%)		2 (2.2%)	
M	M0	420 (79.4%)	209 (79.17%)	211 (79.62%)	0.8822	81 (89.0%)	
	M1	79 (14.93%)	38 (14.39%)	41 (15.47%)		9 (9.9%)	
	Unknow	30 (5.67%)	17 (6.44%)	13 (4.91%)		1 (1.1%)	
N	N0	239 (45.18%)	129 (48.86%)	110 (41.51%)	0.3733	79 (86.8%)	
	N1	16 (3.02%)	11 (4.17%)	5 (1.89%)		2 (2.2%)	
	Unknow	274 (51.8%)	124 (46.97%)	150 (56.6%)		10 (11%)	
Grade	G1	13 (2.46%)	10 (3.79%)	3 (1.13%)	0.0555		
	G2	228 (43.1%)	117 (44.32%)	111 (41.89%)			
	G3	205 (38.75%)	90 (34.09%)	115 (43.4%)			
	G4	75 (14.18%)	41 (15.53%)	34 (12.83%)			
	Unknow	8 (1.51%)	6 (2.27%)	2 (0.75%)			
Stage	Stage I	263 (49.72%)	134 (50.76%)	129 (48.68%)	0.7915		
	Stage II	57 (10.78%)	31 (11.74%)	26 (9.81%)			
	Stage III	123 (23.25%)	58 (21.97%)	65 (24.53%)			
	Stage IV	83 (15.69%)	40 (15.15%)	43 (16.23%)			
	Unknow	3 (0.57%)	1 (0.38%)	2 (0.75%)			

Development and validation of nomogram and PCA analysis

Nomograms were employed to validate the clinical applicability and utility of the risk models. Specifically, nomograms were developed to predict OS at 1, 3, and 5 years by integrating various clinical parameters, including age, gender, pathological stage, histological grade, and risk score (Fig. 4A). Calibration plots were utilized to assess the predictive accuracy of the nomogram prediction model, demonstrating similar trends to the ideal model in predicting patients’ OS at 1, 3, and 5 years, affirming a good fit between the predicted values of the risk model and the actual outcomes (Fig. 4B). Similarly, consistent trends were observed in the TCGA-KIRC testing set, the entire TCGA-KIRC cohort, and the external validation set (Figs. 4C–4H). The PCA scatter plot results across various cohorts indicated the risk model’s ability to distinctly differentiate between low- and high-risk patients (Figs. 4I–4L). Collectively, these indications suggest the accuracy and clinical applicability of the risk model.

Figure 4 Nomograms and PCA analysis of the risk model.

(A and B) Nomograms and calibration plots constructed in the TCGA-KIRC training set to predict OS at 1, 3, and 5 years. (C and D) Nomograms and calibration plots constructed in the TCGA-KIRC testing set to predict OS at 1, 3, and 5 years. (E and F) Nomograms and calibration plots constructed in the entire TCGA-KIRC cohort to predict OS at 1, 3, and 5 years. (G and H) Nomograms and calibration plots constructed in the external validation set to predict OS at 1, 3, and 5 years. (I–L) PCA scatter plots of risk models for high- and low-risk groups in (I) the TCGA-KIRC training set, (J) the TCGA-KIRC testing set, (K) the entire TCGA-KIRC cohort, and (L) the external validation set.

Immune landscape between the low-risk and high-risk groups

Initially, we evaluated the tumor microenvironment within the entire TCGA-KIRC cohort using the ESTIMATE method. The analysis revealed substantially higher estimated and immune scores in the high-risk group compared to the low-risk group (p < 0.001) (Fig. 5A). Subsequently, we employed the CIBERSORT algorithm to determine the distribution of 22 immune cell types within the TCGA-KIRC cohort. The composition of these immune cell types was visualized as a histogram, depicting the relative proportions of distinct immune cells in each sample (Fig. S4). We further utilized the “ggplot2” R package to calculate correlations between risk scores and immune cells. The analysis demonstrated significant positive associations between risk scores and specific immune cell populations, including CD8+T cells, activated CD4+ memory T cells, and Tregs (Fig. 5B). Moreover, we examined the relationship between the expression levels of prevalent immune checkpoint molecules and the risk score. Our findings revealed a robust correlation between the risk score and these immune checkpoints (p < 0.05). Notably, CTLA4, LAG3, TIGIT, PDCD1, SIRPA, TNFRSF9, and LILRB1 exhibited significantly higher expression levels in high-risk KIRC patients compared to low-risk patients (p < 0.001) (Fig. 5C). Additionally, our analysis revealed a correlation between high-risk status and heightened immune-related functions, including APC_co_stimulation, T_cell_co-stimulation, and inflammation promotion (Fig. 5D). Furthermore, we investigated the association between TMB and the risk score. The findings indicated a higher TMB in high-risk patients compared to low-risk patients (p = 0.023). Additionally, we noted that individuals exhibiting both high TMB and high risk displayed the poorest prognosis, whereas those with low TMB and low risk exhibited the most favorable prognosis (Figs. 5E–5G).

Figure 5 Immune landscape and tumor mutation load in low- and high-risk groups of the entire TCGA-KIRC cohort.

(A) Differential analysis of the tumor microenvironment in low- and high-risk groups. (B) Differential analysis of immune cells in low- and high-risk KIRC groups. (C) Differential analysis of immune checkpoints between low- and high-risk groups. (D) Analysis of differences in immune-related functions between low- and high-risk groups. (E) Correlation of risk score with TMB. (F) KM curve showing survival difference between high and low TMB patients. (G) KM curve showing survival difference analysis of patients with combined analysis of TMB and risk score. *p < 0.05; **p < 0.01; ***p < 0.001.

Biological functions of hub genes

To investigate the biological functions of hub genes, we performed a PPI analysis of hub genes and their interacting genes using GeneMANIA, and the network illustrates that the four genes are enriched in DNA damage response, potassium channel complex, cell cycle checkpoint, microtubule polymerization, transmembrane transporter complex, regulation of ion transmembrane transport, detection of chemical stimulus, etc., (Fig. 6A). Then, we further performed GSEA pathway enrichment analysis for each gene in the low and high-risk groups of the entire TCGA-KIRC cohort. In the high-risk group, the top five GO terms were considered to be significantly enriched representatives. The results showed that hub genes were involved in immunoglobulin complex, humoral immune response mediated by circulating immunoglobulin, complement activation, phagocytosis recognition, and humoral immune response (Fig. 6B). The top five KEGG pathways in the high-risk group showed that hub genes were involved in complement and coagulation cascades, cytokine receptor interaction, retinol metabolism, oxidative phosphorylation, and drug metabolism cytochrome p450 (Fig. 6C).

Figure 6 Expression validation and biological functions of hub genes and drug screening.

(A) PPI network presentation of hub genes by GeneMANIA portal. (B) The top five GO terms in the high-risk group. (C) The top five KEGG pathways in the high-risk group. (D) Expression of hub genes in GEO-COVID-19 cohort. (E) Expression of hub genes in TCGA-KIRC cohort. (F–I) Detection of hub gene expression in renal cancer cell lines by qRT-PCR. *p < 0.05, **p < 0.01, ***p < 0.001.

Expression validation of hub genes

Analysis of public databases revealed significant overexpression of hub genes in both COVID-19 and KIRC (Figs. 6D and 6E). Additionally, qPCR results demonstrated variable degrees of overexpression of the four hub genes in RCC cell lines (Figs. 6F–6I). These findings further substantiate the accuracy and reliability of the risk model.

Drug screening

To identify potential drugs for the treatment of COVID-19 and KIRC co-morbidity, we conducted drug target enrichment analysis with four hub genes (GTSE1, CEACAM4, HECW2, and KCNMA1) as genetic targets associated with the construction of risk models via cellMiner and DsigDB databases (Tables S3, S4). The results revealed that Etoposide, Fulvestrant and Topotecan were significantly positively correlated with the expression level of the four hub genes (Table 2). These drugs serve as important references for the development of treatment strategies for comorbidity.

Table 2 Significant drugs of hub genes-related.

Drug	DsigDB.
p-value	CellMiner.
p-value	CellMiner.
Cor	Genes	
Etoposide	0.0096	0.0092	0.3335	CEACAM4, GTSE1	
Methotrexate	0.0104	0.0101	−0.3295	GTSE1, HECW2	
Fulvestrant	0.0388	0.0005	0.4364	CEACAM4, GTSE1	
Vinblastine	0.0446	0.0402	−0.2657	GTSE1, KCNMA1	
Topotecan	0.0491	0.0112	0.3255	KCNMA1	
Melphalan	0.0546	0.0147	0.3135	CEACAM4, HECW2	
Raloxifene	0.0722	0.0003	0.4530	CEACAM4, KCNMA1	
Tamoxifen	0.1510	0.0034	−0.3718	KCNMA1	
Irinotecan	0.2312	0.0191	0.3019	CEACAM4, GTSE1	
Camptothecin	0.2700	0.0099	0.3305	CEACAM4, GTSE1	

Discussion

Although recent research has uncovered the participation of several coronavirus receptor genes in the advancement of KIRC (Choong et al., 2023; Tang et al., 2021; Hossain, Akter & Uddin, 2021; Tripathi et al., 2020), their mechanisms of action remain unclear. Thus, it is significant to evaluate the expression, prognosis, immune function, and other characteristics of COVID-19-related genes in KIRC tissues. These genes can not only function as prognostic biomarkers and drug targets for both COVID-19 and KIRC but also offer novel insights into the underlying pathogenic mechanisms shared by these two diseases.

In this study, we identified 156 shared genes between the COVID-19 and KIRC cohorts and selected four hub genes (GTSE1, CEACAM4, HECW2, and KCNMA1) as the foundation for developing a risk model. By conducting a comprehensive statistical analysis, we validated the model’s accuracy, robustness, and clinical relevance in predicting the prognosis of KIRC. All four genes exhibited notable overexpression in both COVID-19 and KIRC cohorts. Specifically, GTSE1 and CEACAM4 were identified as risk factors for KIRC prognosis, whereas HECW2 and KCNMA1 served as protective factors. GTSE1 is a protein that is specifically expressed during the G2 and S phases of the cell cycle and is involved in the progression and metastasis of various cancers (Monte et al., 2000; Guo et al., 2016; Liu et al., 2019; Xu et al., 2018). Research has demonstrated the significant overexpression of GTSE1 in KIRC tissue. Depletion of GTSE1 expression has been found to suppress the proliferation and invasion of KIRC cells and induce apoptosis both in vitro and in vivo (Chen et al., 2022). Notably, the depletion of GTSE1 leads to the upregulation of KLF4, a tumor suppressor in KIRC (Song et al., 2013). Intriguingly, certain studies have reported that the depletion of KLF4 in lung epithelial cells results in an upregulation of ACE2 expression (Mastej et al., 2022). Therefore, we speculated that there may be a potential connection between GTSE1 and ACE2. CEACAM4, a constituent of the carcinoembryonic antigen family, is frequently overexpressed in cancer and often signifies tumor invasion (Wakabayashi-Nakao et al., 2014). Notably, CEACAM4 exhibits high expression in KIRC and can potentially serve as an independent prognostic marker associated with sunitinib resistance and immune infiltration (Peng et al., 2021). HECW2, a member of the E3 ubiquitin ligase family, plays a pivotal role in kidney development (Qiu et al., 2016). Investigations have revealed the upregulation of HECW2 in colon and cervical cancer, suggesting its potential involvement in tumorigenesis (Lu et al., 2013). KCNMA1, a constituent of the renal K channel, is implicated in the regulation of renal potassium metabolism (Welling, 2016). Extensive research has highlighted the involvement of KCNMA1 in diverse tumor processes. Specifically, it is upregulated in breast cancer, prostate cancer, glioblastoma, and cervical cancer, while downregulated in gastric and colon cancer (Ma et al., 2017; Basile et al., 2019; Du et al., 2016; Bury et al., 2013).

The immune system serves as the primary defense against viral infections, and its decline heightens susceptibility to SARS-CoV-2 (De Biasi et al., 2020). Functional enrichment analysis of shared and hub genes in both diseases reveals significant enrichment of genes associated with immune functions. Immune functional correlation analysis also indicates heightened immune-related functions in the high-risk group of KIRC patients, highlighting the close relationship between shared genes of these diseases and immune function. The high-risk group exhibits notably higher estimated scores and immune scores, with higher estimated scores being linked to lower tumor purity (Yoshihara et al., 2013). Tumors with low purity are associated with a poor prognosis but have a higher likelihood of benefiting from immunotherapy (Gong, Zhang & Guo, 2020). CTLA4, LAG3, TIGIT, PDCD1, SIRPA, TNFRSF9, and LILRB1 are all common and promising immunotherapy targets (Zhou et al., 2020; Chao, Weissman & Majeti, 2012; Li et al., 2020). The more targets there are, the better the expected effect of cancer immunotherapy will be. Therefore, the effect of immunotherapy in high-risk group will be better than that in low-risk group. Comparatively, the high-risk group demonstrates elevated proportions of CD8+ T cells, activated CD4+ memory T cells, and Tregs, which serve as effector T cells, fostering tumor immune activation and cytotoxicity against tumor cells, thereby favoring positive clinical outcomes for patients (Dunn et al., 2002). However, Tregs function as immune suppressive cells that inhibit tumor immune responses and secrete various immune suppressive factors, facilitating tumor immune evasion (Lei et al., 2020). These findings imply that four key genes in KIRC may be involved in immune regulation. The expression levels of immune checkpoints serve as an indicator of tumors’ response to immune checkpoint inhibitors, with overexpression of immune checkpoints leading to immune suppression and evasion (Qin et al., 2019). Our observations reveal significantly higher expression levels of CTLA4, LAG3, TIGIT, PDCD1, SIRPA, TNFRSF9, and LILRB1 in high-risk KIRC patients compared to low-risk patients. Previous studies have established a positive association between higher TMB in tumor cells and increased benefits from immunotherapy (Rizvi et al., 2015). In our study, we found that high-risk KIRC patients exhibited higher TMB compared to their low-risk counterparts. We hypothesize that the poorer prognosis in high-risk patients may stem from immune suppression and evasion triggered by immune checkpoint overexpression. Conversely, high-risk patients may derive greater benefits from immune therapies, including immune checkpoint blockade. These results illustrate the existence of a link between these four hub genes that may be associated with immune function and the predictive potential of immunotherapy effects.

The treatment of patients with both KIRC and COVID-19 presents a challenging task (Ged, Markowski & Pierorazio, 2020). In our study, we identified a total of 3 drugs. Etoposide, a potent topoisomerase II poison, not only functions as an inducer of cancer cell death but also shows promise in treating immune-mediated inflammatory diseases associated with cytokine storm syndrome (Bailly, 2023). Some studies have speculated that etoposide may be a viable option for addressing COVID-19-induced cytokine storms (Delgado-López et al., 2021; Hamizi, Aouidane & Belaaloui, 2020; Lovetrue, 2020). Fulvestrant, a steroid anti-estrogen, has gained approval for treating hormone receptor-positive metastatic breast cancer (Howell, Johnston & Howell, 2004). Research indicates that fulvestrant can significantly inhibit the growth and invasion of RCC cells (Song et al., 2018). Moreover, it enhances the sensitivity of Sunitinib in RCC (Gu et al., 2020), offering a potential new therapy for metastatic RCC (Song et al., 2018). Joshi and Sanjeev et al (Jani et al., 2021; Nayarisseri et al., 2020). Utilized artificial intelligence and machine learning algorithms, concluding that fulvestrant may be an effective treatment for COVID-19. Topotecan, a topoisomerase I (TOP1) inhibitor, has demonstrated efficacy in preventing SARS-CoV-2-induced fatal inflammation (Ho et al., 2021). Furthermore, clinically relevant concentrations of topotecan induce apoptosis more effectively than 5-FU in RCC cell lines (Ramp et al., 2001). Previous studies have demonstrated that the combination of Suberoylanilide hydroxamic acid with topotecan can effectively inhibit the growth of renal cancer cells (Sato et al., 2011). Additionally, topotecan in combination with various drugs may represent a novel treatment for inducing apoptosis in otherwise unresponsive RCC cells (Déjosez et al., 2000; Sato et al., 2009; Sato et al., 2008). These drugs serve as valuable references in the development of treatment strategies for patients simultaneously afflicted with COVID-19 and KIRC.

There are several limitations associated with this study. Firstly, it relies on a retrospective cohort, and the sample size for COVID-19 is relatively small, which could introduce selection bias. To enhance the reliability of our findings, further reinforcement is necessary through future prospective studies. Secondly, the multi-step selection process employed in this study may have overlooked important genes, thereby limiting the applicability of the risk score. Most importantly, the molecular mechanisms underlying the involvement of the four hub genes in KIRC and COVID-19 infection have not been validated in laboratory experiments, warranting further research in the future.

Conclusion

We developed a prognostic risk model based on four co-related genes between COVID-19 and KIRC, which exhibited favorable performance in both the internal and external validation sets. This risk model demonstrated accurate predictive value for the prognosis of KIRC patients and exhibited a strong association with immune function, thereby offering novel insights into the co-morbidity mechanisms of COVID-19 and KIRC. Additionally, we identified several highly correlated drugs through our screening process, potentially providing novel perspectives for the treatment of co-morbidities.

Supplemental Information

Supplemental Information 1 Raw data.

Supplemental Information 2 Heat map showing differentially expressed genes in the COVID-19 cohort.

Supplemental Information 3 Heat map showing differentially expressed genes in the entire TCGA-KIRC cohort.

Supplemental Information 4 Partial results of WGCNA for the COVID-19 cohort of differentially expressed genes.

(A) Demonstration of scale independence and mean connectivity of WGCNA, choosing a soft threshold of 13 and a scale-free topological fit index of 0.9 resulted in relatively balanced scale independence and mean connectivity of WGCNA; (B) Systematic tree diagram and trait correlation heat map of genes.

Supplemental Information 5 Histogram showing the relative proportion of immune cells in high- and low-risk samples.

Supplemental Information 6 Univariate COX regression results for the training set.

Supplemental Information 7 Nine genes generated after LASSO regression.

Supplemental Information 8 Drug analysis results based on cellMiner database.

Supplemental Information 9 Drug analysis results based on DSigDB database.

Supplemental Information 10 MIQE checklist.

All essential information (E) must be submitted with the manuscript. Desirable information (D) should be submitted if available. If using primers obtained from RTPrimerDB, information on qPCR target, oligonucleotides, protocols and validation is available from that source.

Supplemental Information 11 Primer-BLAST Results.

The authors sincerely thank all participants involved in this study.

List of abbreviations

RCC renal cell carcinoma

KIRC kidney renal clear cell carcinoma

SARS-CoV-2 severe acute respiratory syndrome coronavirus 2

ACE2 angiotensin-converting enzyme-2

TCGA The Cancer Genome Atlas

GEO Gene Expression Omnibus

ICGC International Cancer Genome Consortium

FC fold change

FDR false discovery rate

DEGs differentially expressed genes

WGCNA weighted correlation network analysis

LASSO least absolute shrinkage and selection operator

KEGG Kyoto Encyclopedia of Genes and Genomes

GO Gene Ontology

qPCR real-time fluorescent quantitative PCR amplification

ROC receiver operating characteristic curve

AUC the area under the curve

OS overall survival

PCA principal component analysis

TMB tumor mutation burden

PPI protein-protein interaction

GSEA Gene Set Enrichment Analysis

MSigDB Molecular Signature Database

DSigDB Drug Signature Database

KM Kaplan-Meier

Additional Information and Declarations

Competing Interests

Author Contributions

Data Availability

The authors declare that they have no competing interests.

Jianqiang Nie conceived and designed the experiments, performed the experiments, prepared figures and/or tables, and approved the final draft.

Hailang Yang performed the experiments, prepared figures and/or tables, and approved the final draft.

Xiaoqiang Liu analyzed the data, authored or reviewed drafts of the article, and approved the final draft.

Wen Deng conceived and designed the experiments, authored or reviewed drafts of the article, and approved the final draft.

Bin Fu analyzed the data, authored or reviewed drafts of the article, and approved the final draft.

The following information was supplied regarding data availability:

The data is available at TCGA-KIRC; GEO GSE196822, GSE211979; ICGC: ICGC-RECA-EU.

The other datasets generated and analyzed during this study are available in the Supplemental Files.

https://dcc.icgc.org/releases/current/Projects/RECA-EU.

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
