# Peer review of "Identification and validation of shared gene signature of kidney renal clear cell carcinoma and COVID-19"

_PeerJ, doi:10.7717/peerj.16927_

## Round 0.1 · original submission · Major Revisions

Kindly revise in light of reviewer comments.

Reviewer 1 ·

Basic reporting

The study titled "Identification and Validation of Shared Genetic Signature of Kidney Renal Clear Cell Carcinoma and COVID-19" explores the shared gene expression programs between clear cell renal cell carcinoma (KIRC) and COVID-19 and their potential implications for understanding susceptibility and co-pathogenesis.

The quality of the writing of the paper is clear, concise, and easy to understand. The authors use appropriate vocabulary and grammar. The paper is also well-organized and easy to follow.

First of all the title of the study doesn’t reflect the work carried out in the study. Genetic signature sounds like “mutation signature” which is a term often used in cancer genomics thus title should be changed to “gene signature”, “gene expression signature” or something else in that line.

The authors introduce the topic clearly in the introduction, and they provide a clear overview of their methodology in the methods section.

Experimental design

While the study offers valuable insights, there are some limitations and suggestions that should be considered-
In my opinion, they should expand on the analysis and internal and external validation cohorts that they have used in the following text in the last paragraph of the introduction-

Line 88-89 “Using bioinformatics analysis, we constructed a risk model comprising four hub genes. We validated this model using both internal and external validation sets and investigated its role in immune infiltration.”

In lines 204-205, they state: Initially, we conducted differential expression analysis on the COVID-19 dataset to identify differentially expressed genes”. The question is differentially expressed genes between what conditions, and timepoints? They mention it in the methods but they should also mention it here as well to improve readability.

In line 226 “Through this analysis, nine shared genes exhibiting statistical significance were identified” What was the significance cut-off?

In lines 227-228 the authors state “Subsequently, the nine genes underwent further refinement using multivariate Cox regression analysis. Ultimately, four hub genes, namely GTSE1, CEACAM4, HECW2, and KCNMA1, were chosen for constructing the risk model.” What criteria were used to refine and get rid of the 5 genes?

Validity of the findings

The study identifies shared genes between KIRC and COVID-19 and discusses their relevance to immune functions. However, it could benefit from delving deeper into the mechanistic explanations of how these genes contribute to increased susceptibility and co-pathogenesis. Providing a more detailed explanation of the biological processes involved would enhance the understanding of the findings.

The authors mention in Line 359 “These findings underscore the intricate regulation of immune function in KIRC by the four hub genes.” How do they regulate? How do these genes affect the immune functions and tumor microenvironment? The relationship between the four hub genes and immune function is not validated experimentally thus this is an overstatement.


It is important to clarify whether the identified shared genes are causally linked to co-pathogenesis or merely associated with it. Demonstrating causality and validating immune cell type abundance in tumor samples can significantly impact the clinical relevance of the findings.

Similarly here as well in Lines 370-371: “These indications underscore the close connection between hub genes, immune function, and the predictive potential for immunotherapy effectiveness.” The results mentioned merely suggest a potential connection between the immune cells/function and hub genes.

It would be great if authors could validate the identified drugs in the experiments with the cell lines that they already possess, especially NS-1619, Trifluridine, and Mefloquine to see how they affect the expression level of the hub genes and cellular proliferation.

Reviewer 2 ·

Basic reporting

The manuscript is clear and unambiguous, with professional English used throughout. However, minor corrections are required for grammar and readability to further improve the manuscript's quality."

Experimental design

The experimental design is well-structured and sufficient; however, some recommended changes, as outlined in the additional comments, would further enhance the overall robustness and clarity of the study

Validity of the findings

no comment.

Additional comments

1. It is recommended to modify the phrase 'We obtained shared genes between these two diseases based on public databases' to 'We obtained shared genes between these two diseases based on public datasets' for more accurate representation.
2. It is advisable to include specific details about the GEO datasets used, such as the GPL information. Additionally, it would enhance the transparency of the study to provide web access links for all databases mentioned, along with the date of their utilization in the research. This will provide readers with more comprehensive information and facilitate reproducibility of the study.
3. The text mentions the screening of DEGs for COVID-19 using logFC>1 and P-value<0.05 criteria. Considering the common practice in DEGs analysis, it is recommended to also include False Discovery Rate (FDR<0.05) as a criterion for a more stringent control of type I errors, similar to the approach used in the DEGs screening of KIRC data. This would contribute to the robustness of the analysis and align with standard practices in the field.
4. For gene function annotation, in addition to Cluster Profiler, it might be beneficial to utilize online resources such as DAVID or EnrichR. These platforms offer a comprehensive analysis and interpretation of gene functions, pathways, and enrichment that could provide valuable insights complementary to the current analysis.
5. I observed some mistakes in the text. I recommend a thorough review and correction of these errors to enhance the overall clarity and quality of the manuscript
6. I noticed a discrepancy in the figure where the text mentions DEGs screening by p-value for the COVID-19 dataset, but the figure displays adjusted p-values on the y-axis. It would be beneficial to ensure consistency between the text and the figure for clarity and accuracy. Please review and align the information accordingly.
7. Figure 6 and similar figures appear to have reduced visibility. It is recommended to replace these figures with higher-quality versions to enhance clarity and readability

Reviewer 3 ·

Basic reporting

This article is well written and clear on the path with good understanding of the subject matter. Background and introduction of the research is also well conducted. Literature references are provided.

Experimental design

Experiment design is well conducted with sufficient detail and staying in the scope of the research.

Validity of the findings

Results of experiment look promising and provided with sufficient data backing them. Conclusions are well stated and sticking to original research. The research is well conducted and could be helpful in diagnostics of diseases, specifically the bio markers can be handy for research communities.

---

## Round 0.2 · Minor Revisions

Some things need to be addressed in the manuscript before it can be considered for acceptance.

First , several relevant studies need to be cited, e.g.
https://www.hindawi.com/journals/jo/2021/8847307/
Tang Q, Wang Y, Ou L, Li J, Zheng K, Zhan H, Gu J, Zhou G, Xie S, Zhang J, Huang W, Wang S, Wang X. Downregulation of ACE2 expression by SARS-CoV-2 worsens the prognosis of KIRC and KIRP patients via metabolism and immunoregulation. Int J Biol Sci. 2021 May 10;17(8):1925-1939. doi: 10.7150/ijbs.57802. PMID: 34131396; PMCID: PMC8193256.
https://www.frontiersin.org/articles/10.3389/fbioe.2021.744659/full
https://www.frontiersin.org/articles/10.3389/fimmu.2023.1038651/full
Hossain MG, Akter S, Uddin MJ. Emerging Role of Neuropilin-1 and Angiotensin-Converting Enzyme-2 in Renal Carcinoma-Associated COVID-19 Pathogenesis. Infect Dis Rep. 2021 Oct 16;13(4):902-909. doi: 10.3390/idr13040081. PMID: 34698182; PMCID: PMC8544489.

You have mentioned in the discussion 'Consequently, we postulated a potential association between GTSE1 and ACE2, suggesting a plausible link to susceptibility to SARS-CoV-2," but this is absent in the results section. Kindly edit to reflect this.

The results are not well- explained, and more explanation is needed regarding the genes identified in different analyses. For example, "Notably, CTLA4, LAG3, TIGIT, PDCD1, SIRPA, TNFRSF9, and LILRB1 exhibited significantly higher expression levels in high-risk KIRC patients compared to low-risk patients." What are these genes, and what is there significance related to the two conditions?

It is not unexpected that immune-related functions would be common to both conditions. Perhaps the study's significant aspect lies in the discovery of potential treatments. However, it is essential to assess how these compare to existing therapeutics or those suggested in other studies

You have not used DrugBank for correlation between gene and drug. I would suggest that use this to build a consensus alongside DSigDB. If this is not possible, give your reasons.

Expand limitations to include details regarding relevant points above. Detail in silico limitations as well.

Reviewer 2 ·

Basic reporting

The revised version of the article covers all the criteria set forth by the journal.

Experimental design

The revised version of the article covers all the criteria set forth by the journal.

Validity of the findings

The revised version of the article covers all the criteria set forth by the journal.

Additional comments

No comments.

Reviewer 3 ·

Basic reporting

Looks much better after the revisions. No comments from me.

Experimental design

Looks much better after the revisions. No comments from me.

Validity of the findings

Looks much better after the revisions. No comments from me.

---

## Round 0.3 · accepted · Accept

In the abstract, the statement 'Finally, we identified promising drugs for COVID-19 and KIRC,..' is bold. Tone it down to something like 'Finally, we identified potential drugs for COVID-19 and KIRC,..'

The authors have addressed the rest of the points and the manuscript is now acceptable.